# The Quinazoline Otaplimastat (SP-8203) Reduces the Hemorrhagic Transformation and Mortality Aggravated after Delayed rtPA-Induced Thrombolysis in Cerebral Ischemia

**DOI:** 10.3390/ijms23031403

**Published:** 2022-01-26

**Authors:** Hwa Young Song, Jee-In Chung, Angela Melinda Anthony Jalin, Chung Ju, Kisoo Pahk, Chanmin Joung, Sekwang Lee, Sejong Jin, Byoung Soo Kim, Ki Sung Lee, Jei-Man Ryu, Won-Ki Kim

**Affiliations:** 1Department of Neuroscience, Korea University College of Medicine, Seoul 02841, Korea; shy997@korea.ac.kr (H.Y.S.); jeegel@korea.ac.kr (J.-I.C.); jalin7777@gmail.com (A.M.A.J.); cj2013@korea.ac.kr (C.J.); kisu99@korea.ac.kr (K.P.); joungchanmin@korea.ac.kr (C.J.); insu5812@korea.ac.kr (S.L.); holicer@korea.ac.kr (S.J.); 2Research Headquarters, Shin Poong Pharm. Co., Ltd., Ansan 15610, Korea; 135kbs@shinpoong.co.kr (B.S.K.); kslee8808@shinpoong.co.kr (K.S.L.); jeimryu@shinpoong.co.kr (J.-M.R.); 3Institute for Inflammation Control, Korea University College of Medicine, Seoul 02841, Korea; 4Department of Physical Medicine and Rehabilitation, Korea University College of Medicine, Seoul 02841, Korea; 5Department of Anesthesiology and Pain Medicine, Korea University College of Medicine, Seoul 02841, Korea

**Keywords:** otaplimastat, embolic middle cerebral artery occlusion, hemorrhage, recombinant tissue plasminogen activator, matrix metalloprotease, tissue inhibitor matrix metalloproteinase

## Abstract

Acute ischemic stroke is the leading cause of morbidity and mortality worldwide. Recombinant tissue plasminogen activator (rtPA) is the only agent clinically approved by FDA for patients with acute ischemic stroke. However, delayed treatment of rtPA (e.g., more than 3 h after stroke onset) exacerbates ischemic brain damage by causing intracerebral hemorrhage and increasing neurotoxicity. In the present study, we investigated whether the neuroprotant otaplimastat reduced delayed rtPA treatment-evoked neurotoxicity in male Sprague Dawley rats subjected to embolic middle cerebral artery occlusion (eMCAO). Otaplimastat reduced cerebral infarct size and edema and improved neurobehavioral deficits. In particular, otaplimastat markedly reduced intracerebral hemorrhagic transformation and mortality triggered by delayed rtPA treatment, consequently extending the therapeutic time window of rtPA. We further found that ischemia-evoked extracellular matrix metalloproteases (MMPs) expression was closely correlated with cerebral hemorrhagic transformation and brain damage. In ischemic conditions, delayed rtPA treatment further increased brain injury via synergistic expression of MMPs in vascular endothelial cells. In oxygen-glucose-deprived endothelial cells, otaplimastat suppressed the activity rather than protein expression of MMPs by restoring the level of tissue inhibitor of metalloproteinase (TIMP) suppressed in ischemia, and consequently reduced vascular permeation. This paper shows that otaplimastat under clinical trials is a new drug which can inhibit stroke on its own and extend the therapeutic time window of rtPA, especially when administered in combination with rtPA.

## 1. Introduction

Cerebral stroke is the second leading cause of death worldwide, with ischemic stroke accounting for 87% of all cases. Over the past 30 years, all attempts to develop effective cytoprotective agents for cerebral ischemic stroke treatment have failed. The majority of currently available agents present single targets against excitotoxicity, oxidative stress or inflammatory responses occurring during ischemia-reperfusion. Therapeutic failure is potentially attributable to damage or death of brain cells through diverse pathways after ischemia.

Recombinant tissue plasminogen activator (rtPA) is the only agent approved by FDA for patients with acute ischemic stroke. Early post-ischemic treatment of rtPA, for example, within 3–4.5 h after onset, achieves good outcomes [1]. Although controversial, however, the delayed treatment of rtPA might be associated with hemorrhagic transformation and mortality [2,3,4] and has various neurodegenerative effects, including modulation of neuronal excitotoxicity and inflammatory responses [3]. Therefore, only a few patients receive (3~8.5%) and benefit (1~2%) from tPA treatment [5].

Many researchers have shown that tPA has not only thrombolytic activity but also activation of MMPs and other molecular signals [6,7]. These pleiotrophic effects may increase tPA neurotoxicity, worsen edema, and further induce BBB leakage and cerebral hemorrhage [6,7]. Thus, adjunctive therapies targeting MMPs and other tPA-related signals may reduce neurotoxicity of tPA and extend its therapeutic time window in ischemic stroke [8,9,10]. However, the results of clinical trials using available cytoprotectants for combined therapy with rtPA have been unconvincing so far [11].

Previously, we developed otaplimastat (previously designated as SP-8203), a small molecule with a quinazoline-2,4-dione scaffold, as a strong cytoprotectant. We recently completed phase 2 clinical trials [12] and are preparing for a phase 3 trial for otaplimastat in stroke patients. Like other cytoprotective agents, treatment of otaplimastat alone reduced brain damage caused by focal ischemia induced by using a suture-occlusion technique [13,14]. In both in vitro and in vivo experimental ischemia models, treatment of otaplimastat inhibits NMDA receptor-mediated neuronal calcium influx [13] and reduces reactive oxygen species production [14]. 

Due to the continuous failure of cytoprotectants in clinical trials, the use of tPA is currently the standard of care for patients with ischemic stroke. However, tPA exhibits 3 to 4.5 h after ischemia, so its delayed treatment is limited. Therefore, lowering the toxicity of tPA at delayed time after ischemia onset is a very important research task in ischemic stroke. As with most other previously reported cytoprotective agents, otaplimastat was not expected to be effective in clinical trials. Due to ethical issues, FDA recommends that tPA be used together in clinical trials of otaplimastat. In this paper, we studied whether the neuroprotective agent otaplimastat can extend the therapeutic time window of tPA by eliminating the toxicity caused by delayed treatment of tPA. 

In our earlier experiments, we found that otaplimastat reduced cerebral hemorrhagic transformation in focal ischemic brain lesions. In the present study, therefore, we examined the inhibitory effects of otaplimastat on cerebral hemorrhagic transformation and rat mortality using an embolic middle cerebral artery occlusion (eMCAO) rat model susceptible to rtPA thrombolytic therapy. An eMCAO rat model was made by inserting a pre-prepared blood clot. In view of the finding that vascular bleeding after delayed rtPA treatment is closely associated with MMP activation [1,15], we focused on ascertaining whether MMP-related mechanisms are involved in inhibition of rtPA-induced hemorrhagic complications by otaplimastat. 

## 2. Results 

### 2.1. Therapeutic Efficacy 

In general, otaplimastat improved infarct size, edema volume and neurobehavioral score in rats subjected to eMCAO and subsequent rtPA administration (Figure 1a–d). Treatment with rtPA at 6 h after embolism induced a significant increase in cerebral hemorrhagic transformation, which was completely suppressed by otaplimastat (Figure 1e). Similarly, delayed rtPA treatment led to a marked increase in mortality of rats with eMCAO exposure, which was reduced upon pretreatment with otaplimastat (Figure 1f). Physiological variables were maintained within the normal range under all experimental conditions (Appendix A). 

### 2.2. Correlation between MMP Activity and Cerebral Hemorrhage

As shown in Figure 2a, rtPA treatment at 6 h after onset of embolism led to increased hemorrhagic transformation. In ischemic brain tissue, tPA is known to induce BBB leakage through MMP activity and eventually to cerebral hemorrhage [6,7]. To investigate the potential correlations of delayed rtPA treatment with MMP activity and cerebral hemorrhage, we assessed these parameters in brain regions, as shown in Figure 2b. In situ zymography analysis using DQ-gelatin revealed that ischemic insult triggered increased MMP activities in infarct lesions, which were further enhanced by rtPA (Figure 2c). Otaplimastat dramatically blocked MMP activation in infarct lesions of rats exposed to eMCAO/rtPA (Figure 2c,d). Cerebral hemorrhagic transformation was induced by eMCAO and was further aggravated by rtPA treatment. Notably, treatment with otaplimastat led to complete reduction of cerebral hemorrhage in rats exposed to eMCAO/rtPA (Figure 2a,e). Spearman’s correlation analysis revealed a correlation between MMP activity and cerebral hemorrhage size at the ipsilateral side of rat brain (Figure 2f).

### 2.3. Effects of Otaplimastat on MMP and TIMP mRNA Levels in Oxygen-Glucose Derivation (OGD)-Exposed Endothelial Cells

Expression of MMPs is regulated by multiple signaling pathways [16]. To ascertain whether otaplimastat exerts effects on the signaling pathway upstream of MMP expression, we further examined its effects on MMP mRNA levels in OGD-treated endothelial cells, a kind of stroke model that reflects the cellular ischemic injuries in vitro. Previously, ischemic/hypoxic insults have been reported to increase mRNA expression of MMPs, including isotypes 2 and 9, in endothelial cells [17]. Consistent with previous findings, OGD induced elevation of MMP 2 and 9 mRNA (Figure 3a). Interestingly, the mRNA levels of these MMPs were not altered by otaplimastat (Figure 3a). Thus, we focused on whether otaplimastat regulates TIMP, the key endogenous inhibitor of MMPs. OGD treatment led to suppression of TIMP1 and 2 mRNA levels (Figure 3b), which were restored in the presence of otaplimastat (Figure 3c), suggesting that the drug exerts its inhibitory effects on MMP activity via upregulation of TIMP rather than upstream signaling molecules. However, in contrast to TIMP1, TIMP2 protein was not detected under our experimental conditions, probably due to extremely low expression (data not shown).

### 2.4. Effect of Otaplimastat on TIMP Expression in Ischemic Brain Tissue 

Experimental rat models were employed to ascertain whether TIMP1 mRNA and protein levels are similarly regulated by otaplimastat in vivo. MMP 2 and 9 levels in blood and brain lysates were clearly elevated within 1 h after rtPA treatment (7 h after onset of ischemia) (Appendix A). Accordingly, we examined TIMP1 mRNA and protein levels in ischemic rat brains at 1, 6 and 18 h after rtPA treatment. Ischemic insult led to reduced TIMP1 protein expression, which was restored following otaplimastat treatment at all experimental time points (Figure 4a,b). Moreover, regardless of rtPA treatment, otaplimastat restored TIMP1 expression lowered by embolic ischemia (data not shown).

### 2.5. Endothelial Permeability and TIMP1 Expression

In view of the finding that abnormal activity of MMP in cerebral vascular endothelia is associated with increased BBB permeability [18], we examined whether otaplimastat modulates permeability across the endothelial monolayer. Endothelial permeability was increased under OGD conditions and further elevated by rtPA treatment in endothelial cells co-cultured with mixed glial cells (Figure 5a). The increased permeability of endothelial cells exposed to OGD/rtPA was completely blocked by otaplimastat (Figure 5a). TIMP1 levels were decreased (Figure 5b) while MMP activities were increased following combined OGD/rtPA treatment (Figure 5c). Otaplimastat restored the decrease in TIMP1 expression (Figure 5b) and suppressed MMP activities (Figure 5c) induced by OGD/rtPA. However, single cultured endothelial permeability was not affected by OGD/rtPA exposure (Appendix A). The difference in the permeability of single cultured endothelial cells and the permeability of co-cultured endothelia with mixed glia may be influenced by surrounding cells such as glial cells. This hypothesis is supported by a previous study showing that the interaction between astrocyte and endothelial cells affects the permeability of the endothelial blood–brain barrier [19]. 

## 3. Discussion 

In the absence of tPA treatment, the embolic ischemia model is very similar to the permanent ischemia model. Although otaplimastat effectively reduced the intarct size in transient focal ischemic rat models [13,14], it had no protective effect in rats subjected to embolic ischemia. These results are not surprising, as this has been seen with other cytoprotective agents. In our present experiments, however, otaplimastat seems to reduce cerebral infarct size and edema and improve neurobehavioral deficits in rats subjected to eMCAO and delayed rtPA treatment. In particular, otaplimastat markedly reduced intracerebral hemorrhagic transformation and mortality triggered by delayed rtPA treatment, consequently extending the therapeutic time window of rtPA. These results indicated that otaplimastat effectively inhibits adverse effects of rtPA treatment at 6 h after ischemia which is defined as “delayed rtPA treatment” in this study. The beneficial outcomes with otaplimastat are not attributable to effects on rtPA-mediated thrombolytic activity (Appendix A). Furthermore, otaplimastat was found to be safe when co-administered with other widely used antiplatelets such as aspirin and clopidogrel. Thus, while Aspirin and clopidogrel extended tail bleeding time, otalplimastat did not alter the tail bleeding time in single or combined treatment with those antiplatelets (Appendix A). 

Several MMP family members play important roles in development, injury and repair in neurological diseases [20]. MMP inhibitors inactivate functional MMPs by binding specific sites at zinc domains [21]. MMPs play a role in normal physiological processes including tissue morphogenesis, cell migration and angiogenesis. MMPs are also involved in pathophysiological processes including wound healing, inflammation and cancer. Due to the destructive and protective actions of MMPs, however, their irreversible inhibition may lead to unexpected side-effects in clinical trials. For this reason, many previous MMP inhibitors exert adverse effects, such as musculoskeletal pain, inflammation and other complications, in clinical studies that are not observed in preclinical experiments [22], which thus limit their use in the clinic [23]. 

Although controversial, MMPs may allow infiltrating cells, including leukocytes, metastatic and transformed cells, to penetrate ECM barriers. In cerebral ischemia, the increase of BBB permeability is mediated by activation of matrix metalloproteinase (MMP), and especially MMP-9 [24,25]. When administered with r-tPA, specific MMP-9 inhibitors markedly reduced brain hemorrhage, swelling, infarction, disability and death, suggesting that blocking the deleterious effects of MMP-9 may improve outcomes after ischemic stroke [26]. Previously, several researchers studied how tPA can regulate MMP. In rtPA-administered ischemic stroke patients, Gołąb and his coworkers revealed that plasmin, which is known to be activated by rtPA, increases MMP-3 activity and influences the subsequent activation of MMP-9 [27]. Moreover, on OGD treated bEND3 cells, Song et al. deciphered that the activation of MMP-2/9 following rtPA treatment is regulated by nitric oxide dependent Cav-1 S-nitrosylation and its downstream ERK signaling pathway [28].

Data from the present study indicate that otaplimastat regulates MMP activity rather than expression. MMP expression increased in cultured vascular endothelial cells treated with OGD. Notably, otaplimastat did not suppress mRNA or protein levels of MMP-2 and-9 in OGD-stimulated cultured endothelial cells (Figure 3) but restored TIMP1 mRNA and protein expression suppressed by OGD. Based on these findings, we propose that otaplimastat reduces MMP activity by upregulating TIMP1 rather than directly inhibiting MMP expression. Otaplimastat showed little toxicity in phase 1 and 2 clinical trials (data not shown). The low toxicity of Otaplimastat may be related to inhibition of MMP activity via TIMP rather than directly inhibiting MMP, but more research on this is needed.

Four endogenous TIMPs, designated TIMP1~4, have been identified to date. The natural ratio of MMPs to TIMPs is tightly regulated and disruption in this balance is often associated with progression of multiple disease states. Each of the four TIMPs forms a complex with MMPs in a 1:1 stoichiometry with high affinity but varying degrees of selectivity [29]. TIMP1 has been suggested as an alternative to overcome the toxicities of direct MMP inhibitors [30,31]. Previous studies have demonstrated lower BBB permeability and infarction following cerebral ischemia in TIMP1-overexpressing transgenic mice relative to their wild-type counterparts [32]. Additionally, direct application of recombinant TIMP1 or in vivo delivery via adenoviral vectors induced a significant reduction in ischemic damage [31,33]. However, exogenous TIMP1 cannot be administered, since it has a short half-life in vivo and is unable to cross the BBB [30,31]. To our knowledge, no synthetic low molecular weight TIMP1 inhibitors have been developed to date. Therefore, inactivation of MMP by otaplimastat through endogenous regulation of TIMP1 may present a safer and more effective therapeutic strategy against cerebral ischemic injury. 

Brain injury after ischemia is induced via activation of various pathways, including excitotoxicity, radical formation and inflammatory responses [34]. In recent years, attempts to develop multi-target-directed drugs for stroke treatment have been made [35,36,37]. Similar to ischemia/hypoxia, rtPA thrombolysis itself induces excitotoxicity, radical formation and inflammatory responses [3] and also exacerbates ischemia/hypoxia-evoked neurodestructive activities [38] as well as cerebral hemorrhage through modulation of vasoactivity and proteolysis affecting the neurovascular matrix [39,40]. Therefore, the optimal therapeutic strategy for acute ischemic stroke is to simultaneously block the common pathways of brain damage caused by ischemia and delayed rtPA treatment. Previously, we reported that otaplimastat exerts mild anti-excitotoxic and anti-oxidative activities [13,14]. In line with the present findings, these data may support protective effects of otaplimastat on brain tissue in stroke patients receiving rtPA.

Many neuroprotective agents have been studied as adjunctive therapies to reduce the risk of cytotoxicity, intracerebral hemorrhage and reperfusion injury of rtPA, and to extend its therapeutic time window. However, to date, few neuroprotective agents have been shown to be effective in clinical trials. In the present study, otaplimastat reduced cerebral ischemic injury and improved neurobehavioral deficits. In addition, otaplimastat significantly reduced intracerebral hemorrhagic transformation and mortality triggered by delayed rtPA treatment. Recently, we reported that intravenous otaplimastat adjunctive therapy in patients receiving rtPA was feasible and generally safe [12]. Our present findings may lead to a better understanding of the pharmacological efficacy and action mode of otaplimastat in combination therapy with tPA. In addition, our present sudy will be of great help in conducting clinical trials with more patients in phase 3 clinical trials. 

## 4. Materials and Methods 

### 4.1. Reagents and Antibodies 

Otaplimastat was chemically synthesized by ShinPoong Pharm. Co., Ltd. (Ansan, Korea). Tissue plasminogen activator (rtPA; Actilyse^®^) was obtained from Boehringer Ingelheim (Ingelheim am Rhein, GE). 2,3,5-Triphenyltetrazolium chloride (TTC) was purchased from Alfa Aesar (Haverhill, MA, USA) and 4′,6-diamidino-2-phenylindole (DAPI) and DQ^TM^ gelatin fluorescein conjugate from Thermo Fisher (Waltham, MA, USA). Other chemicals, including Drabkin’s reagent, were acquired from Sigma-Aldrich (St. Louis, MO, USA). 

### 4.2. Cell Culture

The mouse brain endothelial cell line (bEnd.3, ATCC CRL-2299) was purchased from American Type Culture Collection (ATCC; Manassas, VA, USA) and grown in Dulbecco’s modified Eagle’s medium (DMEM; Welgene, Korea) supplemented with 10% FBS (Hyclone, Logan, UT, USA) and 1% penicillin-streptomycin (Hyclone, Logan, UT, USA) at 37 °C in a humidified incubator with 95% air and 5% CO_2_. For mixed glia cultures, cerebral cortices were isolated from Sprague-Dawley rat brains at postnatal day 1–2. Cortices were mechanically dissociated from meninges and triturated gently with a flame-polished Pasteur pipette in culture medium. Cells were plated at a density of 5 × 10^5^ cells/mL in Minimum Essential Medium (Hyclone, Logan, UT, USA) supplemented with heat-inactivated 10% fetal bovine serum (Hyclone, Logan, UT, USA), 2 mM glutamine (Hyclone, Logan, UT, USA) and 1% penicillin-streptomycin. Plates were coated with poly-d-lysine (2 μg/mL, Sigma-Aldrich, St. Louis, MO, USA) for 2 h. Cells were incubated at 37 °C in a humidified incubator with 95% air and 5% CO_2_, and 4 d later, half of the culture medium was replaced with a fresh medium. Mixed glia were used at 7–8 d after culture.

### 4.3. Oxygen-Glucose Deprivation (OGD)

To mimic ischemic conditions in vitro, cells were exposed to OGD as described previously [41]. Briefly, medium was replaced with glucose-free DMEM (Sigma-Aldrich, St. Louis, MO, USA) and cells incubated in an anaerobic chamber (partial pressure of oxygen < 2 mmHg) for 6 h at 37 °C. Otaplimastat was added at the same time as OGD induction. 

### 4.4. Animal Models

Adult Sprague-Dawley (SD) male rats (260–300 g) were purchased from Charles River Laboratories (Wilmington, MA, USA) and maintained under a 12 h light/dark cycle with ad libitum access to food and water. Acclimated rats were initially anesthetized with 3% isoflurane in a N_2_O and O_2_ (7:3, *v*/*v*) mixture and maintained with 2% isoflurane. Body temperature was maintained at 37 ± 0.3 °C and monitored using a rectal thermometer. All experimental protocols and procedures were approved by the Ethics Committee and the Institutional Animal Care and Use Committee of Korea University College of Medicine (Approval No. KOREA-2018-0030). All experiments were performed in accordance with the approved guidelines and regulations. The detailed methodology is described in the checklists of the animal research reporting in vivo experiments (ARRIVE) guideline (Appendix A).

### 4.5. Embolic Middle Cerebral Artery Occlusion (eMCAO) 

The eMCAO rat model was generated according to a previously described method with specific modifications [42]. Briefly, a blood clot (emboli) was generated from donor rat femoral arterial blood using polyethylene (PE) tubing (PE-50, ID: 0.58 mm, OD: 0.99 mm, Intramedic, DK). PE-50 tubing containing donor blood was incubated for 2 h at room temperature (20 to 25 °C) followed by incubation at 4 °C for 22 h. The clots in PE-50 tubing were subsequently flushed out from the tubing by a syringe with normal saline and cut in small pieces, of which length is 3.5~4 cm, in a Petri dish. Then, one of the clot pieces was selected and transferred into a modified PE-50 catheter (OD: 0.72~0.75 mm) for clot injection. Under a surgical microscope (Carl Zeiss, Inc., Thornwood, NY, USA), the modified PE-50 catheter containing the blood clot was inserted into the external carotid artery (ECA) and advanced to 17 mm from bifurcation between ECA and internal carotid artery (ICA). Next, the clot was gently injected with normal saline into ICA using a 100 μL Hamilton syringe after which the catheter was carefully removed. The sham control experiment followed the same surgical procedure without clot injection. Rats subjected to MCAO with emboli were randomly assigned to individual groups after examining for signs of stroke (e.g., forelimb flexion with spontaneous circling). Rats with no evidence of stroke or seizure after stroke induction were excluded for further experiments.

### 4.6. Drug Administration 

Otaplimastat (3 mg/kg) was dissolved in normal saline and administered intravenously through the right femoral vein at 4.5 and 5.25 h after eMCAO. Subsequently, rtPA (10 mg/kg) was intravenously infused via the right femoral vein at 6 h after eMCAO. A proportion (10%) of the total rtPA volume was administered as bolus and the remaining volume administered over 30 min.

### 4.7. Measurement of Infarction and Edema 

Infarct and edema volumes were measured as described previously [43]. After completion of all experimental procedures, rats were deeply anesthetized with 3% isoflurane and transcardially perfused with cold normal saline for removal of whole body blood. Rat brains were extracted, placed in a rat brain matrix (Ted Pella, Redding, CA, USA), and cut into 2 mm coronal sections. Brain sections were stained with a 2% TTC solution at room temperature for 15 min. The infarct volume was measured using an image analysis program, TOMORO ScopeEye version 3.5 (Techsan Digital Co., Seoul, Korea). Edema volume was determined based on the formula: edema volume (%) = [(V_I_ – V_C_)/V_C_] × 100, whereby V_I_ represents the ipsilateral hemisphere volume (affected hemisphere) and V_C_ the contralateral hemisphere volume (unaffected hemisphere). As the observed infarct volume could be overestimated due to brain edema, total corrected infarct volume was quantified by integrating all brain sections after compensating for brain edema [44]. Corrected infarct volume (IV_C_, mm^3^) was calculated as IV_d_ × (V_C_/V_I_), whereby IV_d_ represents ipsilateral infarct volume obtained by direct measurement. The percentage of infarct volume was calculated against whole brain volume (infarct volume (%) = IV_C_/V_W_ × 100, whereby V_W_ represents whole brain volume). 

### 4.8. Measurement of Neurobehavioral Deficits 

Each animal was assigned a score on the neurological scale [45], specifically, 0 = no deficit, 1 = failure to extend left forelimb, 2 = decreased grip of the left forelimb if pulled by the tail, 3 = spontaneous movement in all directions, contralateral circling if pulled by the tail, 4 = circling or walking to the left, 5 = movement only when stimulated and 6 = unresponsive to stimulation. 

### 4.9. Measurement of Intracerebral Hemoglobin Concentrations

Intracerebral hemorrhage is quantified based on an optical spectrophotometric assay assuming the level is proportional to the amount of hemoglobin [46]. To standardize the hemoglobin level, brain hemispheres of naïve rats were subjected to transcardial perfusion with cold normal saline to remove whole body blood and extracted. Incremental volumes of homologous blood (0, 2, 4, 8, 16, 32 µL) were added to each hemispheric sample with saline to reach a total volume of 1.5 mL followed by homogenization for 30 s, sonicated on ice for 1 min and centrifuged at 18,000× *g* for 30 min. In total, 800 μL Drabkin’s reagent (Sigma-Aldrich, St. Louis, MO, USA) was added to 200 μL supernatant and allowed to stand for 15 min at room temperature. Optical density was recorded at 540 nm with a spectrophotometer (SpectraMAX190; Molecular Devices, San Jose, CA, USA). We observed a linear relationship between hemoglobin concentrations in perfused brain and volume of added blood. Measurements from perfused brains were compared with this standard to obtain hemoglobin concentration data (μL). Microscopic hemorrhage size was calculated as reported previously [47].

### 4.10. RNA Isolation and Real-Time Quantitative Reverse Transcription PCR

To quantify tissue inhibitor of metalloproteinase (TIMP)-1 and -2 mRNA expression levels, total RNA was extracted from bEnd.3 cells with the TRIzol RNA extraction kit (Thermo Fisher Scientific, Waltham, MA, USA). Total RNA (1 μg) was reverse-transcribed to cDNA using SuperScript IV VILO Master Mix (Bio-rad, Hercules, CA, USA). For quantitative analysis, reverse transcription PCR was performed on a CFX96 Touch™ Real-Time PCR Detection System (Bio-rad, Hercules, CA, USA) using iQ^TM^ SYBR^®^ Green Supermix (Bio-Rad, Hercules, CA, USA). The primers used were TIMP1 sense 5′ TGTGGGAAATGCCGCAGATA 3′ and antisense 5′ GGCCCGTGATGAGAAACTCT 3′, TIMP2 sense 5′ CCACAGACTTCAGCGAATGG 3′ and antisense 5′ AGAGGAAGGCAGCAATGACT 3′, MMP2 sense 5′ CCCTGGTGCTCCACACTTCA 3′ and antisense 5′ TGTATCCCACTGCCCTGTGC, MMP9 sense 5′ CTTCACCTTCGAGGGACGCT 3′ and antisense 5′ CGTTGCCATGCTCCGTGTAG, GAPDH sense 5′ AAGGCTGTGGGCAAGGTCAT 3′ and antisense 5′ TTTCTCCAGGCGGCATGTCA 3′. 

### 4.11. Enzyme-Linked Immunosorbent Assay (ELISA)

Cultured supernatant was immediately collected after OGD stimulation and incubated at −80 °C for further experiments. Total TIMP1 and TIMP2 levels were quantified in 96-well microtiter plates using commercially available ELISA kits (RayBiotech, Norcross, GA, USA) according to the manufacturer’s recommendations.

### 4.12. Western Blot

Brain tissues were collected at 7, 12 and 24 h after eMCAO onset and homogenized in RIPA buffer (Thermo Fisher Scientific, Waltham, MA, USA) supplemented with a proteinase inhibitor cocktail (GenDEPOT, Katy, TX, USA). The supernatant was collected after centrifugation at 13,000× *g* for 15 min at 4 °C and protein concentrations determined with BCA reagent (Thermo Fisher Scientific, Waltham, MA, USA). Each protein sample (15 μg) was denatured by boiling in Laemmli buffer (93.75 mM Tris-Cl (pH 6.8), 12.5% glycerol, 2.25% SDS, and 2.25% beta-mercaptoethanol), separated on a 12% SDS polyacrylamide gel and transferred onto polyvinylidene fluoride membrane (PVDF, Bio-Rad, Hercules, CA, USA). Membranes were incubated for 1 h in TBS-T (Tris-buffered saline and 0.1% Tween 20) containing 5% skimmed milk and incubated with antibodies against TIMP1 (1:200; Santa Cruz Biotechnology, Santa Cruz, CA, USA), ZO-1 (1:500; Invitrogen, Carlsbad, CA, USA) and anti-β-actin (1:1000; Cell Signaling, Danvers, MA, USA) overnight, followed by incubation with horseradish peroxidase (HRP)-conjugated secondary antibody (1:5000; Thermo Fisher Scientific, Waltham, MA, USA). Immunoreactivity was detected using Amersham ECL^TM^ Prime Western Blotting Detection Reagent (GE healthcare, Chicago, IL, USA). 

### 4.13. Permeability Assay

The integrity of the endothelial barrier was evaluated using the lucifer yellow (LY) permeability assay to assess permeability across the endothelium monolayer, based on a modified version of the published protocol [48]. To establish the blood–brain barrier (BBB) endothelium model, bEnd3 cells were plated onto a 24-transwell plate with polyester membrane inserts (Corning, NY, USA) at a seeding density of 5 × 10^4^ cells/cm^2^. The transwell membrane was coated with fibronectin (30 μg/mL; GenDEPOT, Katy, TX, USA) for 2 h at 37 °C and washed with PBS. For cell seeding, 250 μL cell suspension was added to the upper chamber and 1 mL fresh medium to the lower chamber. Cells were incubated at 37 °C in a humidified incubator with 95% air and 5% CO_2_ for 9–10 days and the medium changed every 2 days. 

On the experimental day, bEnd3 cells were transferred to plates of mixed glial cells. Cell plates were treated with otaplimastat and placed in an anaerobic chamber (partial pressure of oxygen < 2 mmHg). To terminate OGD, 2.5 mM glucose was added and 20 μg/mL rtPA administered. Cells were incubated at 37 °C in a humidified incubator under 95% normal air and 5% CO_2_. After 4 h, supernatant fractions were collected and stored at −80 °C for TIMP1 ELISA and zymography. To assess permeability, 250 μL transport buffer (4 mM KCl, 141 mM NaCl, 2.8 mM CaCl_2_, 1 mM MgSO_4_, 10 mM HEPES, and 10 mM d-glucose, pH 7.4) containing 100 μM LY was added to the upper chamber and 1 mL fresh transport buffer to the bottom chamber, followed by incubation of the transwell plate for 1 h at 37 °C in a humidified incubator. LY concentrations in the bottom chamber were determined using Spectra MAX GEMINI EM (Molecular Devices, San Jose, CA, USA) at excitation and emission wavelengths of 470 and 539 nm, respectively. The apparent permeation coefficient (*Pc*) was calculated using the following equation: $$Pc\;\left(\mathrm{cm}/\min\right)=\frac{V_{b}\times C_{b}}{C_{u}\times A\times T}$$
where *V_b_* is the volume of the bottom chamber (1000 μL), *C_b_* the concentration of LY (μΜ) in the bottom chamber, *C_u_* the concentration of LY (μΜ) in the upper chamber, *A* the membrane area (0.33 cm^2^) and *T* the time of transport (60 min). 

### 4.14. Gelatin Zymography

Gelatin zymograms were used to assess MMP-2 and MMP-9 levels in ischemic brain homogenates and supernatants of cultured endothelia or mixed glia. The supernatant fractions (500 μL) were concentrated via centrifugation at 21,000× *g* for 30 min at 4 °C using Centrifugal Concentrator VIVASPIN 500 (10 kDa; Sartorius, Göttingen, GE). Concentrated supernatants and 50 μg of each brain homogenate were mixed with non-denaturing sample buffer (93.75 mM Tris-Cl (pH 6.8), 12.5% glycerol and 2.25% SDS) and separated on a 10% SDS polyacrylamide gel containing 0.1% gelatin as substrate. The gel was incubated with zymogram renaturing buffer (Novex, Carlsbad, CA, USA) for 1 h at room temperature followed by zymogram development buffer (Novex, Carlsbad, CA, USA) for 24 h at 37 °C. To visualize protein bands, the gel was stained with the Colloidal blue staining kit (Invitrogen, Carlsbad, CA, USA). 

### 4.15. In Situ Zymography 

In situ zymography was applied to examine MMP activation in ischemic brain sections using a fluorescent DQ-substrate (Invitrogen, Carlsbad, CA, USA) as described previously [49]. At the desired time point, extracted brain tissues were immediately embedded using OCT compound and frozen on dry ice. Embedded brains samples were sectioned (8 to 20 μm) using a cryostat and air-dried for 1 h at room temperature, rehydrated with PBS and incubated with 20 μg/mL DQ-gelatin substrate in zymogram developing buffer (Invitrogen, Carlsbad, CA, USA) at 37 °C overnight. After three washing steps in PBS, mounted slides were examined under a confocal microscope (LSM 510, Zeiss, Oberkochen, GE) to detect green fluorescence due to gelatinolytic activity. MMP activity was quantified as fluorescence intensity using Image J software (National Institute of Health, Bethesda, MD, USA). 

### 4.16. Statistical Analysis

All analyses were performed using IBM SPSS statistics 20 software (IBM, Armonk, NY, USA). Data were expressed as median ± standard deviation (SD) or standard error of the mean (SEM) and median ± interquartile ranges from Q1 to Q3. Statistical significance was assessed with one-way analysis of variance (ANOVA) followed by post-hoc Tukey’s test or Games-Howell test for multiple comparisons or by using the Kruskal–Wallis test followed by Mann–Whitney test. 

## Figures and Tables

**Figure 1 ijms-23-01403-f001:**
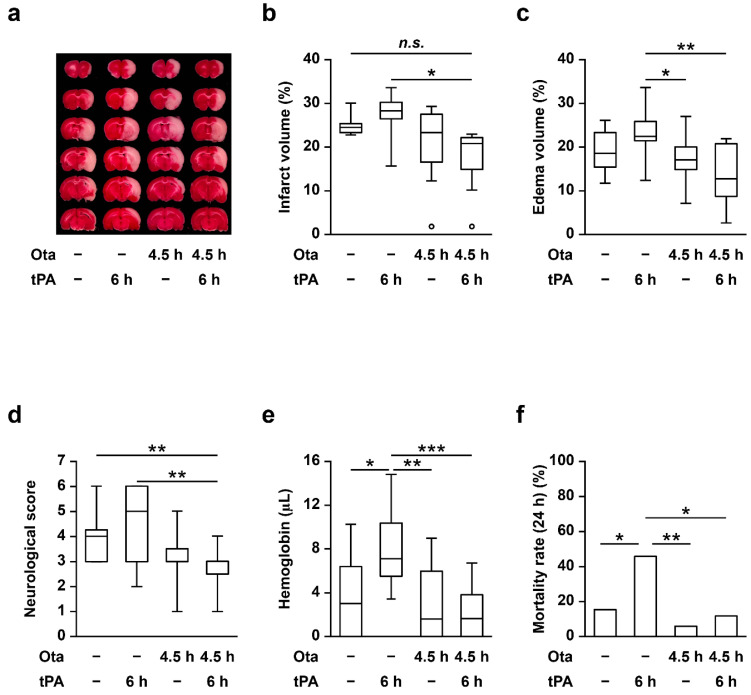
Effects of otaplimastat (Ota) on pharmacological efficacy, vascular bleeding and mortality in rats subjected to eMCAO and rtPA. Otaplimastat (3 mg/kg) and rtPA (10 mg/kg) were intravenously administered at 4.5 and 6 h after the onset of eMCAO, respectively. Representative TTC-stained brain slices (**a**), infarct volume (**b**), edema volume (**c**), neurological deficit score (**d**), hemoglobin influx in the ischemic brain (**e**) and mortality rate (**f**) were assessed at 24 h after eMCAO onset. Except for hemorrhage and mortality, error bars represent median ± interquartile range (Q1 to Q3) and were analyzed via one-way ANOVA followed by post-hoc analysis with Tukey test (*n* = 9–11). The hemorrhage and mortality rates were analyzed with Kruskal–Wallis followed by Mann–Whitney test (*n* = 17–26, * *p* < 0.05, ** *p* < 0.01 and *** *p* < 0.001, ° indicates outlier).

**Figure 2 ijms-23-01403-f002:**
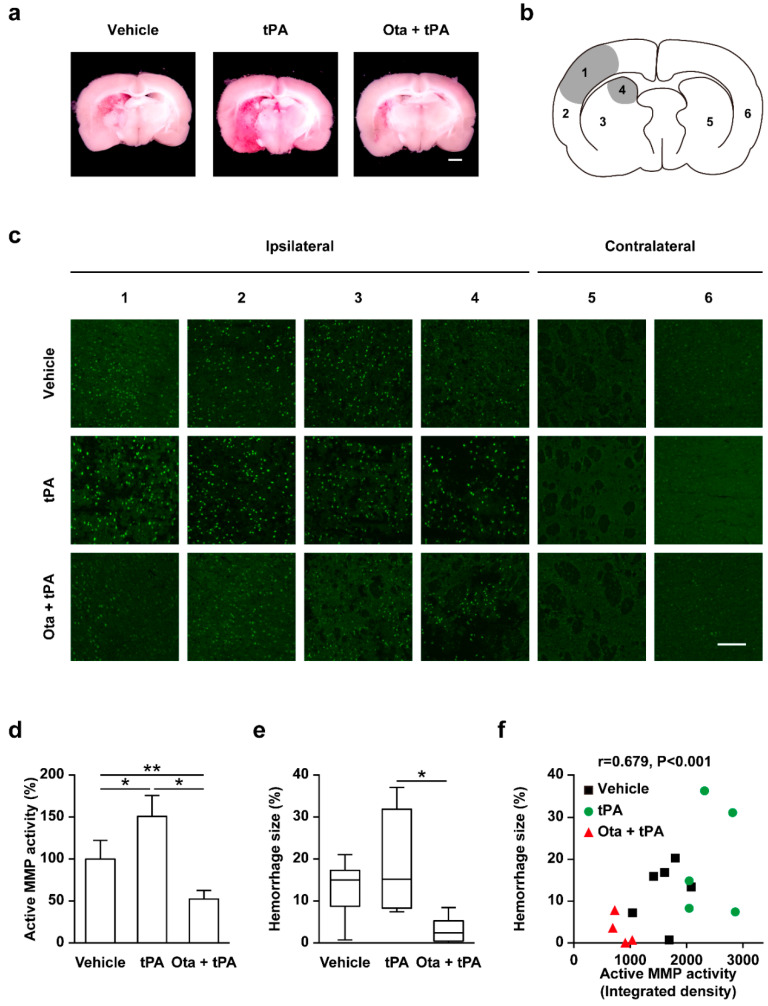
Reduction of MMP activity and hemorrhage size by otaplimastat (Ota). (**a**) Representative brain slices (scale bar = 100 μm). (**b**) Schematic representation of whole brain. (**c**) Representative in situ gelatin zymography of the area shown in (**b**). Quantification of MMP activity (**d**) and microscopic hemorrhage (**e**) measured 7 h after embolism in the vehicle, rtPA and otaplimastat/rtPA groups, and correlation between MMP activity and hemorrhage volume (**f**). Quantified data are expressed as means ± S.D. Microscopic hemorrhage volumes are expressed as median ± interquartile rages (Q1 to Q3). Statistical differences were analyzed with Kruskal–Wallis followed by Mann–Whitney test, * *p* < 0.05 and ** *p* < 0.01. Correlation between MMP activity and hemorrhage volume determined using the Spearman’s test (R = 0.679, *p* < 0.001; *n* = 4–6 per group).

**Figure 3 ijms-23-01403-f003:**
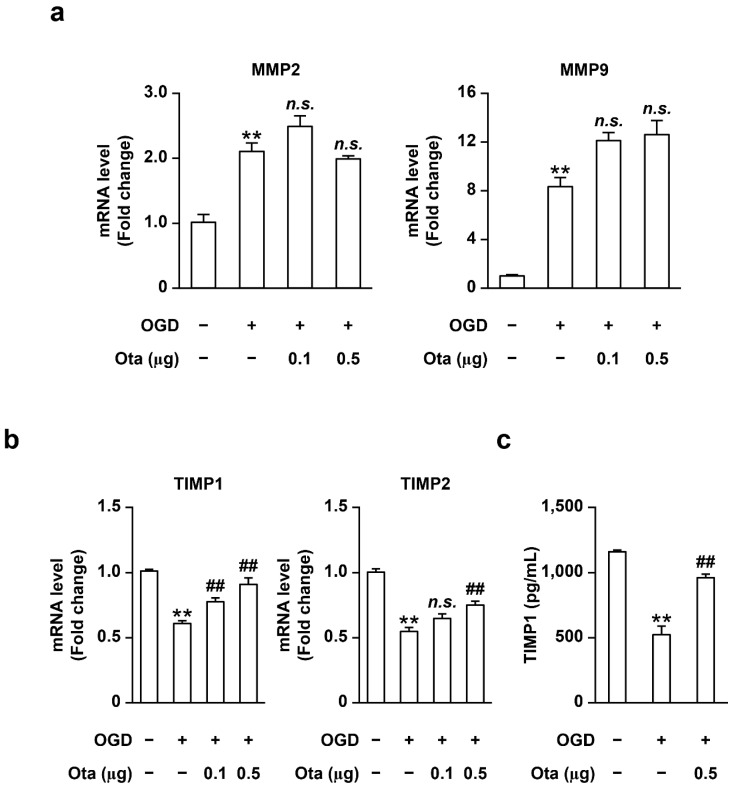
Effects of otaplimastat (Ota) on MMP and TIMP mRNA expression in bEND3 cells. (**a**) MMP-2 and MMP-9 mRNA levels. (**b**) TIMP1 and TIMP2 mRNA levels. (**c**) TIMP1 protein levels in the supernatant. Experimental bEnd3 cells were treated with OGD for 6 h in the absence or presence of otaplimastat (0.1 and 0.5 μM). Data are presented as means ± S.E.M and analyzed with Kruskal–Wallis followed by Mann–Whitney test (** *p* < 0.01 vs. normoxia, ^##^
*p* < 0.01 vs. OGD treated group).

**Figure 4 ijms-23-01403-f004:**
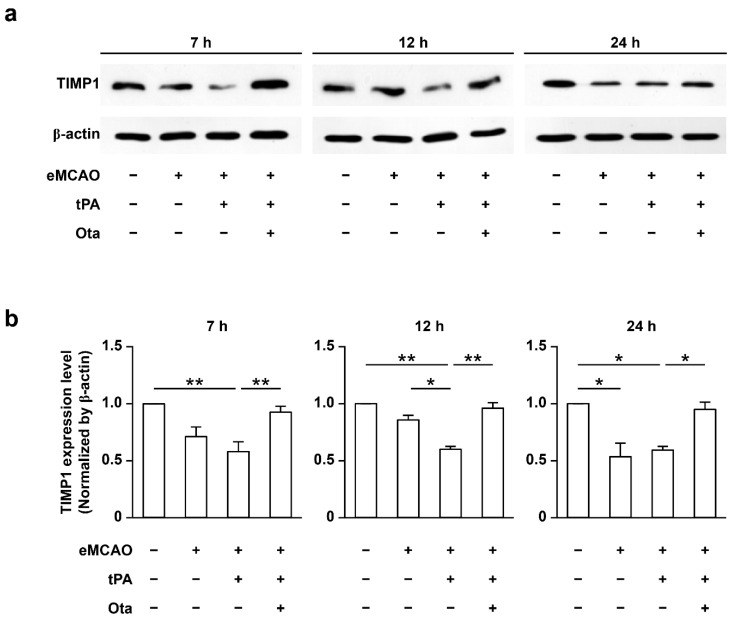
Restoration of TIMP1 levels in embolic brains by otaplimastat (Ota). (**a**) Rats were treated with rtPA at 6 h after eMCAO onset and TIMP1 levels determined via Western blot at 7, 12 and 24 h. (**b**) Integrated density values of TIMP1 protein levels were normalized to that of β-actin. Data are expressed as means ± S.D. and analyzed with Kruskal–Wallis followed by Mann–Whitney test (*n* = 6, * *p* < 0.05 and ** *p* < 0.01).

**Figure 5 ijms-23-01403-f005:**
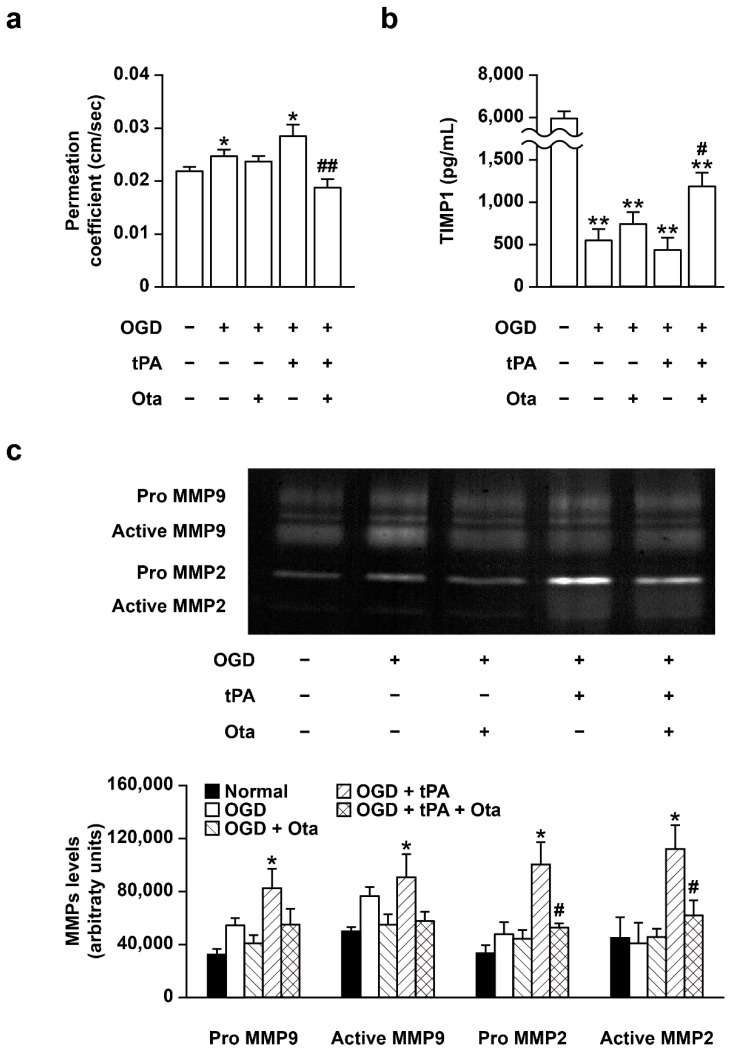
Otaplimastat (Ota) inhibits endothelial vascular permeability enhanced by OGD/rtPA treatment in endothelial bEnd3 cells co-cultured with mixed glia. bEnd3 cells co-cultured with mixed glia in transwell plates were subjected to OGD in the absence and presence of otaplimastat (10 μM). Cells were treated with rtPA 2 h after the onset of OGD. (**a**) Trans-endothelial permeability measured via lucifer yellow flux as optical density (O.D.). (**b**) TIMP1 expression in bEnd3 cells cultured on the upper chamber. (**c**) Representative zymogram and quantification of MMPs in supernatant fractions. Otaplimastat inhibited rtPA-induced MMP activation. Data are expressed as means ± S.D. and statistical differences analyzed with Kruskal–Wallis followed by Mann–Whitney test (*n* = 3–6, * *p* < 0.05 and ** *p* < 0.01 versus normal control group, ^#^
*p* < 0.05 and ^##^
*p* < 0.01 versus OGD + rtPA group).

## Data Availability

The data presented in this study are available on request from the corresponding author.

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
