# Peer review of "The Quinazoline Otaplimastat (SP-8203) Reduces the Hemorrhagic Transformation and Mortality Aggravated after Delayed rtPA-Induced Thrombolysis in Cerebral Ischemia"

_ijms, 2022, doi:10.3390/ijms23031403_

Round 1

Reviewer 1 Report

Dear authors, I see that you have qualified the anti-stroke effects of otaplimastat by itself with the following sentence in the Introduction:

Like other cytoprotective agents, treatment of otaplimastat alone reduced brain damage caused by focal ischemia induced by using a suture-occlusion technique (Noh et al., 2011, a and b). 

You also dealt with the problem by writing in the discussion :

Although otaplimastat effectively reduced the intarct size in transient focal ischemic rat models (Noh et al., 2011, a and b), it had no protective effect in rats subjected to embolic ischemia. 

Yours sincerely

Author Response

thanks

Reviewer 2 Report

There does not seem to be a clear effect of Ota. To substantiate the authors' claim, two-way ANOVA should be performed. It is actually confusing that tPA & Ota dual treatment is similar to non-treated control.

Reviewer 3 Report

This study tested studied Otaplimastat as a neuroprotectant in a preclinical setting (in-vivo experiments and cell cultures) of large vessel stroke and delayed lysis therapy.

The study is very complete, methodologically extremely cohesive and over all scientifically sound. I would like to congratulate particularly for your figures and figure descriptions, which are very thoroughly arranged and well explained. I think this study merits publication.

However, there are some issues that should be addressed first:

  • The parts of the animal experiments needs a more detailed methodological description. I struggle to understand the real study design. How many animals were operated? How many died prematurely? Ideally you add a flowchart showing exact animal numbers per treatment arm. Furthermore, I ask you to strictly adhere to ARRIVE guidelines. In order to avoid misunderstandings, you may upload a set of the ARRIVE guidelines with the revised manuscript, indicating where in the manuscript you report the relevant item.

  • I miss a clear conclusion. There is no conclusion indicated and in your last paragraph of the discussion section, you discuss some of your unpublished data. Please add a designated conclusion paragraph.

  • With the paragraph on mechanical thrombectomy you are leaning quite far out of the window of a highly controversial topic. Just backing up with “unpublished data” is not strong enough for your point here. Either present these data, or revise the paragraph.

  • It is a bit odd to begin the Abstract with a self-reference. Please revise.

  • The second sentence of the Abstract (Recombinant tissue plasminogen activator 31 (rtPA) is the only agent clinically approved by FDA for patients) needs embedding in context. Obviously, there are many agent approved by the FDA for patients, but probably not for stroke lysis.

  • Page 4 line 64ff: Clinically speaking, the risk of delayed tPA is a revascularization syndrome. Here, you are directly starting on a molecular level (with an accurate and completely true description). However, I suggest you begin more clinically orientated, given that you want to describe the sequel of a delayed clinical therapy.

  • Page 10, line 133: It took me a while to realise that you describe all of sudden a complete other set of experiments (in-vitro) instead of continuing with the in-vivo part. Please make that clearer for the reader already early on.

  • Page 16, line 206: ASA and clopidogrel are thrombocyte-aggregation inhibitors and not (plasmatic) anticoagulants. I think the term anticoagulants is a bit misleading here.

  • Page 16 line 210: This is quite random a change of the subject form one paragraph to the next. I do not see why all of a sudden you start discussing MMPs, after having concluded that Otaplimastat works well with ASA and clopidogrel. Please revise to guide the reader better into the depths of the topic of MMPs.

Author Response

This manuscript is a resubmission of an earlier submission. The following is a list of the peer review reports and author responses from that submission.

Round 1

Reviewer 1 Report

In general, the outcome of otaplimastat (Ota) administration does not appear to be definitive. Results described for Figure 1 needs to be evaluated by two-way ANOVA to test the main effect of Ota. For most results, it appears that tPA is detrimental and Ota improves the tPA-induced detrimental effects; however, the outcome of tPA+Ota appears to be not greatly different from non-treated control. Therefore, the readers will be left confused about whether Ota is therapeutically useful.

Author Response

Responses to the reviewer #1’s comments:

General comment. Re: (1) In general, the outcome of otaplimastat (Ota) administration does not appear to be definitive. (2) Results described for Figure 1 needs to be evaluated by two-way ANOVA to test the main effect of Ota. For most results, it appears that tPA is detrimental and Ota improves the tPA-induced detrimental effects; however, the outcome of tPA+Ota appears to be not greatly different from non-treated control. Therefore, the readers will be left confused about whether Ota is therapeutically useful.

  1. Re: definitive outcome.

Answer: The purpose of the present study was to see the therapeutic effect of otaplimastat on eMCAO/tPA-evoked brain damage. Thus, the experimental design was employed to see the drug effect of maximally suppressing brain damage caused by eMCAO and tPA in combination, not eMCAO alone. Brain ischemic lesion is divided into two lesions: penumbra vs. core. Unlike ischemic penumbra lesion, the core lesion is irreversibly damaged. Therefore, all stroke drugs are being developed for the purpose of inhibiting the penumbra, which accounts for up to 20% of the total brain (i.e., 40% of brain hemisphere), as measured by TTC staining. Since the experimental conditions applied in this study already induce maximum brain damage, it is not possible to see much larger increase after tPA treatment, but the inhibitory effect of the used drug can be observed. Instead, tPA aggravates cerebral hemorrhage and mortality (fig. 1, e and f). Therefore, 15-20% of brain damage suppression can be recognized as a very good drug effect. It is rare for other researchers to report a drug with an inhibitory effect beyond the inhibitory effect we obtained in this study. Thus, we think that the therapeutic effect of otaplimastat is good enough.

  1. Re: statistical analysis

Answer: As the reviewer comments, two-way ANOVA could be better for two variables as used in this experiment (fig. 1). However, both one-way and two-way ANOVA can be used for multiple statistical comparison, because we fixed variables. In addition, both statistical analytic results were the same. The researcher who directly conducted the experiment is currently at a new job in the Netherlands, so re-analysis of the data was not made in the manuscript.

Reviewer 2 Report

The comments are also presented throughout the text as sticky pads.  In the uploaded file.

In short:  This is a very interesting paper, which however needs a little more background, and various clarifications throughout ( in Abstract, Introduction, Results, Discussion, and Methods )

First of all, I would start the title with :  The Quinazoline Otaplimastat (SP-8203)…….

I think it would get the paper to a running start by catching the interest of at least two more groups of readerships.

Individual comments  ( indicated by sticky pads throughout the ms.  here and there are also textual corrections, in blue):

Line 29

I think this is an overstatement.  Your results show an effect on Neurological Deficit Score (Fig. 1) but on one of the other parameters, compared to untreated stroke.

Did you check regeneration ?

Line 30

Present the cell culture results here also, including full name of OGD

Line 43

Could you please say something about perinatal stroke and advanced age stroke ?

I only recently realized that perinatal stroke is a very serious problem, and I guess that like me, not sufficient people are aware of this tragic prevalence of perinatal stroke.  At least as advanced as advanced age stroke.

Line 44

Could you say something about the mechanisms whereby rtPA ameliorates the effects of stroke ?  It is an important piece of information, as a handful of other agents have effects at least as good on stroke as rtPA ( Veenman, 2020 ).

It would be interesting to know whether the various efficacious agents all operate via the same sets of molecular biological mechanisms .

For example, MMP modulation may cause side effects in all of the alternative treatments.

Lines 45 and 49

rtPA

Line 50

functions of MMPs are described in more detail in the Discussion.

Line 52

Could you please say how MMPs exert their effects and how they are affected by rtPA " i.e. the molecular biological mechanisms involved ?

Line 58

Would SP-8203 bind to TSPO?  Various quinazolines bind to TSPO and have ameliorating effects on stroke (Chen et al., 2017).  In general, ligands for TSPO have ameliorating effects on stroke ( Dimitrova-Shumkovska et al., 2020)

Line 65

Rat mortality

Line 66

Say here that you cause the emboli by inserting previously prepared blood clots

Line 85

Only here ( Neurological Deficit Score )  Ota compares favorable to untreated. ( see my note on "overstatement" in the Abstract.

Line 90

Symbol not exactly as in figure.

Line 113

d,e,f are not marked in the figure.

please have arrows or arrow heads point at the labeled cells in c

Line 118

what is OGD ?

Line 122

not clear what you mean with "upstream" here. Please give more detail of what you are thinking about.

Line 127

in contrast to TIMP1, TMP2

Line 179

present ? previous ?

Lines 186 – 189

This sentence needs more background and explanation.  Possibly to include it into the main part of the paper, including the figure.

Line 224

Veenman, 2020

Line 242

question is here " what happens at cellular, histological, and neuroanatomical levels. And more details of molecular biological mechanisms "

Line 261

brains i.e. plural ?

Line 271

again, also here no description.

" What is OGD ? "   ( I am not going to Google it ! )

Line 302

So, permanent stroke. No reperfusion, or clot removal. Please mention shortly that it is permanent.

Line 304

emphasize this ( in normal saline ), various drugs targeting the brain are dissolved in hydrophobic vehicles, which often have toxic effects.

Line 322

Everybody is happy with these measurements and calculations ?

Edema makes the infarct look bigger ( not relatively smaller ? )

I am just curious.  Please discuss this a bit.

Line 414

please express in g ( not in rpm )

Line 450

OGD is also not here.

Early in the text ( with the first abbreviation ) say " see list of abbreviations "

as lists go, the list is not complete.  I would say, either include all the abbreviations used in the text ( including for example the name of the cells used in the cell culture experiments ) or do not present this list.  For most part, I had no problems with the abbreviations used in the text. ( except for OGD, ah, found it, finally Oxygen-glucose deprivation )

Author Response

Responses to the reviewer #2’s comments:

Soon after I uploaded my manuscript, I re-uploaded my new version. I communicated with Ms. Zhu about the new version. Unfortunately, many of reviewer 2's comments were based on the earlier version of the article. I think the new version was not delivered to the reviewer #2 in time. However, referring to reviewer2's comments, I tried to correct as much as possible.

Comment 1. Re: the comments presented throughout the text as sticky pads in the uploaded file.

Answer: All comments were answered and corrected directly in the text.

Comment 2. Re: the title change.

Answer: As the reviewer suggested, the title is started with “The Quinazoline Otaplimastat (SP-8203)…….” in the revised manuscript (line 1)

Comment 3. Re: Line 29. I think this is an overstatement. Your results show an effect on Neurological Deficit Score (Fig. 1) but on one of the other parameters.

Answer: The purpose of the present study was to see the therapeutic effect of otaplimastat on eMCAO/tPA-evoked brain damage. In the absence of tPA, embolic MCAO continuously damages the brain, causing more than 40% of rats to die within 3 days after eMCAO onset (data not shown in the manuscript). Rats subjected to eMCAO and tPA both died more than 80%, in which we cannot perform the experiments. Thus, the present experimental design was employed to see the drug effect of maximally suppressing brain damage caused by eMCAO and tPA in combination, not eMCAO alone. Therefore, to examine the effect of otaplimastat on tPA treatment, the test was performed on the 1st day after ischemia. Because eMCAO alone already submaximally induces infarction and edema, treatment with tPA at 6h after eMCAO onset does not increase brain damage by more than 10%. Thus, the most important comparison for therapeutic effect is tPA group (not untreated group) vs. otaplimastat/tPA group. As all readers see, infarct volume, edema volume, neurological score, hemorrhage and mortality rate in rats treated with tPA for 6 h were reduced by otaplimastat treatment.

Comment 4. Re: Line 30. Present the cell culture results including full name of OGD.

Answer: In response to the reviewer’s comment, the cell culture results including full name of OGD are newly stated in the revised manuscript (lines 38 and 39).

Comment 5. Re: Line 43. Could you please say something about perinatal stroke and advanced age stroke? I only recently realized that perinatal stroke is a very serious problem, and I guess that like me, not sufficient people are aware of this tragic prevalence of perinatal stroke.  At least as advanced as advanced age stroke.

Answer: In case of peripheral stroke, drugs with a cytoprotective effect will have a therapeutic effect. Since otaplimastat has a cytoprotective effect, I think it will be effective to some extent. However, in the case of advanced age stroke, there is a high possibility of cerebral hemorrhagic bleeding with ischemia, so otaplimastat is thought to have a particularly good therapeutic effect. Further studies are needed in these models. However, this statement is not included in the revised manuscript.

Comment 6. Re: Line 44. Could you say something about the mechanisms whereby rtPA ameliorates the effects of stroke?  It is an important piece of information, as a handful of other agents have effects at least as good on stroke as rtPA (Veenman, 2020). It would be interesting to know whether the various efficacious agents all operate via the same sets of molecular biological mechanisms. For example, MMP modulation may cause side effects in all of the alternative treatments.

Answer: Despite previous reports showing anti-ischemic effects in stroke, all neuroprotective reagents have been failed in clinical trials. Thus, rtPA is the only drug approved by FDA for the treatment of embolic cerebral ischemia. tPA dissolves blood clots and improves blood flow. However, delayed treatment of tPA exacerbates the outcome of embolic stroke. Many mechanisms for the detrimental effects of rtPA are the induction of oxidative stress, the recruitment of inflammatory cells into ischemic lesion, etc. tPA also increases MMP levels and activities in endothelial cells and induces leakage of blood brain barrier. These are partially mentioned in the Introduction section. 

Comment 7. Re: Lines 45 and 49. rtPA

Answer: Following the reviewer’s instruction, I corrected the spelling in the revised manuscript (lines 37, 42, 60 and 64).

Comment 8. Re: Line 50. functions of MMPs are described in more detail in the Discussion.

Answer: In response to the reviewer’s comment, the function of MMPs are stated in more detail in the Discussion section (lines 157-160, 164-169) in the revised manuscript.

Comment 9. Re: Line 52. Could you please say how MMPs exert their effects and how they are affected by rtPA " i.e. the molecular biological mechanisms involved ?

Answer: In response to the reviewer’s indication, we discussed how tPA regulates MMP in the revised manuscript with reference to previously reported clinical and non-clinical studies (lines 169-175)

Comment 10. Re: Line 58. Would otaplimastat bind to TSPO?  Various quinazolines bind to TSPO and have ameliorating effects on stroke (Chen et al., 2017).  In general, ligands for TSPO have ameliorating effects on stroke (Dimitrova-Shumkovska et al., 2020)

Answer: The reviewer’s comment is very interesting and well worth researching. However, the interaction of otaplimastat with TSPO was not investigated in this study.

Comment 11. Re: Line 65. Rat mortality

Answer: Since it was mentioned in the sentence that a rat model was used, I did not redundantly state that the death was about the rat.

Comment 12. Re: Line 66. Say here that you cause the emboli by inserting previously prepared blood clots

Answer: According to the reviewer’s comment, it was stated in the last part of Introduction section that an embolic MCAO rat model was made by inserting a pre-prepared blood clot. (line 80)

Comment 13. Re: Line 85. Only here (Neurological Deficit Score) Ota compares favorably to untreated. (see my note on "overstatement" in the Abstract.

Answer: This comment is basically the same with comment 3. As afore-mentioned in the comment 3, brain damage in the untreated group is already submaximal. Thus, in the present experimental condition rtPA does not further increase the infarct volume, edema volume, neurolofgical score and hemorrhage. The main onjective of this study was to see whether otaplimastat improved each index in tPA-treated rats.

Comment 14. Re: Line 90. Symbol not exactly as in figure.

Answer: As I mentioned, this comment is based on the wrong earlier version of the article. The new version was not timely delivered to the reviewer. In the new version of the manuscript, the symbols in figures and text correctly match.

Comment 15. Re: Line 113. d,e,f are not marked in the figure. please have arrows or arrow heads point at the labeled cells in c

Answer: This comment is also based on the wrong earlier version of the article. In the new version of the manuscript, the labels and symbols in figures and text correctly match.

Comment 16. Re: Line 118. what is OGD?

Answer: The abbreviation of OGD is already defined in line 105. Thus, the full name of OGD is not repeatedly stated

Comment 17. Re: Line 122. not clear what you mean with "upstream" here. Please give more detail of what you are thinking about.

Answer: The protein expression and activity of MMP can be regulated at the level of mRNA and/or by an independent regulatory protein TIMP. Because mRNA level was not changed by otaplimastat, we mentioned that the upstream signaling for MMP mRNA expression is not affected by otaplimastat. Rather, othaplimastat regulated the expression of TIMP, consequently affecting the MMP activity. However, to avoid reader’s confusion, we erased this sentence in the revised manuscript.

Comment 18. Re: Line 127. in contrast to TIMP1, TIMP2

Answer: According to the reviewer’s comment, ‘in contrast to TIMP1,’ was inserted before ‘TIMP2’ in the revised manuscript (line 117)

Comment 19. Re: Line 179. present? previous ?

Answer: The word ‘present’ is mentioned in the revised manuscript (line 145)

Comment 20. Re: Lines 186 ~189. This sentence needs more background and explanation.  Possibly to include it into the main part of the paper, including the figure.

Answer: In general, anti-coagulants with a mechanism different from that of tPA may enhance or sustain the antithrombotic effect of tPA. Therefore, it is essential to study the interaction of these drugs with tPA. We studied the interaction of tPA with aspirin or clopidogrel, the most commonly used two anti-coagulants in hospitals. However, the result was shown only as an supplementary materials because it is separate from the flow of the main content.

Comment 21. Re: Line 224. Veenman, 2020

Answer: In response to the reviewer’s indication, the Veenman’s paper was referenced in the revised manuscript (lines 198, 575 and 576)

Comment 22. Re: Line 242. question is here " what happens at cellular, histological, and neuroanatomical levels. And more details of molecular biological mechanisms "

Answer: A more detailed experimental results obtained at cellular, histological, neuroanatomical levels and molecular biological mechanism would improve the quality of our manuscript. However, those results require time for research and are too much to be included in this paper. Thus, the answers to the reviewer’s valuable comments may be published in a separate article.

Comment 23. Re: Line 261. brains i.e. plural?

Answer: In response to the reviewer’s indication, ‘brain’ was replaced with ‘brains’. (line 231)

Comment 24. Re: Line 271. again, also here no description. " What is OGD?  

Answer: The abbreviation of OGD is already defined in line 105. Thus, the full name is not repeatedly stated

Comment 25. Re: Line 302. So, permanent stroke. No reperfusion, or clot removal. Please mention shortly that it is permanent.

Answer: Embolic stroke cannot be induced in all rats by inserting blood clot. Ischemia is induced in as much as 80% of the rats used in this study. Therefore, rats that were not properly induced to ischemia were no longer used for further experiments such as treatments of tPA and/or otaplimastat. To avoid the reader’s confusion, ‘from analysis’ was replaced with ‘for further experiments’. (line 269)

Comment 26. Re: Line 304. emphasize this (in normal saline), various drugs targeting the brain are dissolved in hydrophobic vehicles, which often have toxic effects.

Answer: Dissolving the drug substance in saline to make an injection has many advantages. Less toxicity after injection is one important advantage. In fact, otaplimastat itself is quite hydrophobic and not easily dissolved in water. For the clinical trials, however, Shinpoong Pharmaceutical corporation makes raw materials into very fine powders through a special method and freeze-dried them to dissolve well in saline. Nevertheless, I do not think it is correct to emphasize that otaplimastat is dissolved in normal saline solution. This is simply because many other drugs are also soluble in physiological saline solution.

Comment 27. Re: Line 322. Everybody is happy with these measurements and calculations? Edema makes the infarct look bigger (not relatively smaller?) I am just curious.  Please discuss this a bit.

Answer: The ischemic region expands due to edema. Therefore, since it is larger than the actual brain volume, it should be corrected. Since this method is widely used by other researchers, detailed description will degrade the quality of the paper.

Comment 28. Re: Line 414. please express in g (not in rpm)

Answer: As the reviewer indicates, ‘rpm’ was replaced with ‘g’ unit in the revised manuscript. (line 373)

Comment 29. Re: Line 450. OGD is also not here.

Answer: The abbreviation of OGD is already defined in line 108. Thus, the full name od OGD is not repeatedly stated

Round 2

Reviewer 1 Report

I am confused.

Reviewer 2 Report

This is what the other reviewer said.  ( I basically said the same in my report. )

"For most results, it appears that tPA is detrimental and Ota improves the tPA-induced detrimental effects; however, the outcome of tPA+Ota appears to be not greatly different from non-treated control. Therefore, the readers will be left confused about whether Ota is therapeutically useful."

Right!

Presently, the authors bent over backward to argue that Otaplimastat is good for otherwise untreated stroke.  To ameliorate damage.  Maybe so, but indeed, just maybe.

Where the authors want to state these believes, they should say :  " It may be assumed " 

Where reviewers have such question marks, any critical reader will have the same question marks.  This does weaken the paper considerably.  That Otaplimastat ameliorates detrimental effects of rtPA is already good enough, and that it has a positive effect on neurological score is also very interesting.  In the end that is what one wants :  that normal behavior is preserved ( motor performance, sensory input ? cognition ? )

Presently, the authors go one bridge too far.  With too little information.

Thus, my binding advice is that anytime when the authors state this point, they have to say " It may be assumed "  

For the rest, the revised manuscript is okay.